# Failure to Rescue (FTR) and Pitfalls in the Management of Complex Enteric Fistulas (CEF): From Rescue Surgery to Rescue Strategy

**DOI:** 10.3390/jpm12020292

**Published:** 2022-02-16

**Authors:** Stefano Piero Bernardo Cioffi, Osvaldo Chiara, Luca Del Prete, Alessandro Bonomi, Michele Altomare, Andrea Spota, Roberto Bini, Stefania Cimbanassi

**Affiliations:** 1General Surgery and Trauma Team, ASST GOM Niguarda, Piazza Ospedale Maggiore 3, 20162 Milan, Italy; stefanopiero.cioffi@ospedaleniguarda.it (S.P.B.C.); osvaldo.chiara@unimi.it (O.C.); michele.altomare@ospedaleniguarda.it (M.A.); andrea.spota@ospedaleniguarda.it (A.S.); roberto.bini@ospedaleniguarda.it (R.B.); 2Department of Pathophysiology and Transplants, State University of Milan, Via Festa del Perdono 7, 20122 Milan, Italy; 3General and Liver Transplant Surgery Unit, Fondazione IRCCS Ca’ Granda, Ospedale Maggiore Policlinico, 20122 Milan, Italy; luca.delpre90@gmail.com; 4General Surgery Residency Program, State University of Milan, Via Festa del Perdono 7, 20122 Milan, Italy; alessandro.bonomi@unimi.it; 5Department of Surgical Sciences, Sapienza University of Rome, Piazzale Aldo Moro 5, 00185 Rome, Italy

**Keywords:** complex enteric fistulas, rescue surgery, rescue strategy, failure to rescue, general emergency surgery, integrated management

## Abstract

Purpose: Complex enteric fistulas (CEF) represent general surgeons’ nightmare. This paper aims to explore the impact on failure-to-rescue (FTR) rate of a standardised and integrated surgical and critical care step-up approach. Methods: This was a retrospective observational cohort study. Patients treated for CEF from 2009 to 2019 at Niguarda Hospital were included. Each patient was approached following a three-step approach: study phase, sepsis control and strategy definition phase, and surgical rescue phase. Results: Sixteen patients were treated for CEF. Seven fistulas were classified as complex entero-cutaneous (ECF) and nine as entero-atmospheric fistula (EAF). Median number of surgical procedures for fistula control before definitive surgical attempt was 11 (IQR 2–33.5). The median time from culprit surgery and the first access at Niguarda Hospital to definitive surgical attempt were 279 days (IQR 231–409) and 120 days (IQR 34–231), respectively. Median ICU LOS was 71 days (IQR 28–101), and effective hospital LOS was 117 days, (IQR 69.5–188.8). Three patients (18.75%) experienced spontaneous fistula closure after conversion to simple ECF, whereas 13 (81.25%) underwent definitive surgery for fistula takedown. Surgical rescue was possible in nine patients. Nine patients underwent multiple postoperative revision for surgical complications. Four patients failed to be rescued. Conclusion: An integrated step-up rescue strategy is crucial to standardise the approach to CEF and go beyond the basic surgical rescue procedure. The definition of FTR is dependent from the examined population. CEF patients are a unique cluster of emergency general surgery patients who may need a tailored definition of FTR considering the burden of postoperative events influencing their outcome.

## 1. Introduction

Acute care surgeons (ACS) deal daily with surgical complications secondary to elective and emergency abdominal surgery for both traumatic and non-traumatic diseases. The ability of ACS to deliver urgent and tailored care is essential to surgical rescue. Some authors have recently published papers addressing the importance of integrating rescue surgery as a pillar of acute care surgery, trauma, emergency surgery, and critical care [1,2,3,4,5,6]. 

One of the most dreaded complications of surgical rescue procedures in such patients is the development of complex enteric fistula (CEF), including complex enterocutaneous fistula (cECF) and enteroatmospheric fistula (EAF). cECF is defined as the absence of criteria for favourable outcomes and loss of the surrounding skin and soft tissue. EAF is an abnormal communication between the enterocolic tract and atmosphere, such as that occurring in grade 4 open abdomen according to the Bjork classification [7].

Acute care surgeons treating patients with CEF encounter serious challenges because the aberrant connection between the bowel and the atmosphere is only the tip of the iceberg. Anatomical alteration is the expression of an underlying altered pro-inflammatory state leading to metabolic, nutritional, and inflammatory deregulation [8,9].

Surgery for CEF is one of the best examples of surgical rescue.

Most patients undergo more than one procedure before the development of CEF. Moreover, most of the rescue procedures are performed in low-volume and low-performance hospitals before referral to high-volume centres. It is difficult to rescue patients with CEF because surgical factors are not the critical ones for definitive care. The contributing factors to rescue the patient embrace a multidisciplinary team, including nutrition, infectious disease, critical care, and psychology professionals and implementation of different bedside surgical techniques for CEF control [3,6,7,8].

In European settings, the absence of structured acute care surgery and surgical critical care programs discloses a potential pitfall in the management of critically ill patients undergoing surgery. Implementation of multidisciplinary pathways is mandatory. In this setting, trauma and emergency surgeons should play a pivotal role to improve patient-centred multidisciplinary teams.

Failure-to-rescue (FTR) is defined as reported mortality after a complication. Silber et al. defined the metric in 1992 as an indicator for patients undergoing elective surgery. Recently, it has been applied to patients with trauma and those undergoing emergency surgery. However, it is unclear whether the definition of FTR should consider surgical and medical complications, as it depends on the population being studied. 

Considering the underlying pro-inflammatory storm and related metabolic deregulation, the approach to CEF should go beyond the simple surgical procedure to rescue the patient. Different surgical bedside approaches have been described for CEF control as a bridge to definitive surgery or for non-operative resolution, while the patient undergoes nutritional and metabolic optimization. A definitive surgical repair should be attempted when the patient is well nourished, and septic sources have been controlled. The surgical planning includes a defined approach for bowel resection with subsequent possible anastomosis and abdominal wall reconstruction. Each plan should be tailored to the patient’s anatomy [7,8,9,10].

Considering CEF as a complication, the FTR rate reached 70% in the past decades, which recently fell to <20% owing to intensive care and surgical improvements [10,11,12]. 

A defined rescue strategy should be implemented to achieve the best metabolic conditions for definitive care. This strategy should consider nutritional support, control of enteric spillage, and sepsis in addition to a proper reconstructive surgical strategy. A standardised multidisciplinary approach is crucial to overcome the concept that FTR in patients with CEF is mainly due to the patient’s baseline condition, leading to limited opportunities for care.

In this study, we reviewed our experience in managing CEF during the last 10 years and evaluated the FTR rate after implementation of a standardised and integrated surgical critical care approach.

## 2. Methods 

We performed a retrospective observational cohort study.

Patients with CEF who were managed at the General Surgery–Trauma Team Unit at Niguarda Hospital, Milan, from 2009 to 2019 were considered. 

Inclusion criteria: patients affected by complex enterocutaneous fistula or enteroatmospheric fistula.

Exclusion criteria: patients affected by simple enterocutaneous fistula with favourable criteria for nonoperative resolution as history of appendicitis or diverticulitis, transferrin >200 mg/dL, no obstruction, length of fistula >2 cm, output <200 mL/24 h, no sepsis, no electrolyte disturbances, and early referral to tertiary care centre. 

Our centre is a level 1 trauma centre for the metropolitan area of Milan, delivering trauma and acute surgical care to 3.25 million people. Our system follows the structure of U.S. level 1 trauma centres thanks to a historical collaboration with R. Adams Cowley Shock Trauma Centre in Baltimore, Maryland. Each patient was managed according to a step-up approach based on three phases: study phase, sepsis control and strategy definition phase, and surgical rescue phase. 

Data on demographics, clinical history, surgical history, type of EF, diagnostic approach, management of EF, P-Possum score [13] for postoperative risk estimation, attempt for definitive surgical treatment, postoperative complications, and outcomes were recorded. Patients’ information were anonymized and collected in a retrospective online database and stored in a computerized spreadsheet (Microsoft Excel 2016; Microsoft Corporation, Redmond, WA, USA).

Data were reported according the Strengthening the Reporting of Observational Studies in Epidemiology guidelines [14] for observational studies.

Descriptive statistics were calculated. Continuous variables are presented as medians and interquartile ranges (IQRs) or means and standard deviations, and categorical variables are presented as numbers and percentages.

## 3. Results 

During the study period, 16 patients, with a median age of 46 years (IQR 36.6–59 years), were treated by our General Surgery–Trauma Team for CEF.

### 3.1. Study Phase 

#### Centralization of Care, Clinical History, and Patients’ Study 

All patients were referred to our hospital from peripheral low-volume centres to rescue complications.

Most patients (56.3%) were men. The median age at admission to our centre was 46 years (IQR 36.6–59 years). Patient clinical and surgical histories before the occurrence of CEF are shown in Table 1. The median number of procedures before the occurrence of CEF was 5.5 (IQR 3–6.3).

The median number of fistulas per patient was 2 (IQR 1–3), totalling to 27. There were seven patients with cECF and nine patients with EAF.

In all patients, the anatomy of the fistulas was studied using contrast-enhanced investigations. The location of fistulas and type of radiological evaluation performed are summarized in Table 2. 

In total, 68% of patients had medium- or high-output fistulas.

### 3.2. Sepsis Control and Strategy Definition Phase

#### Bedside Dressing, Endoscopic Approach, and Metabolic Support

For each patient, at least two different dressing techniques were combined to optimize output control and wound care. Whenever a new dressing technique was implemented or a significant revision was necessary, the procedure was performed in the operating room (OR); otherwise, dressings were changed bedside at least once a day. The median number of procedures for fistula control performed in the OR was 11 (IQR 2–33.5).

The nipple technique or the Fistula Adapter™ (Phametra, Pharma & Medica-Trading GmbH, Herne, Germany) technique was used at least once in 62.5% cases, while the floating stoma technique was used at least once in 50% cases (Table 3).

In 81.25% cases, negative pressure wound therapy (NPWT) was applied in association with other dressings to promote granulation of the tissues surrounding the fistula. Once achieved, the granulation tissue was covered with a split-thickness skin graft to allow the placement of an ostomy bag over the graft in 31.25% cases.

In seven patients (43.75% cases), the effluent control was optimized, and we were able to convert the CEF to a simple stoma. In three of these patients (42.8%), the fistula healed spontaneously with no need for a definitive surgical approach. 

Nine patients (56.3%) underwent endoscopy to improve fistula output control. Endoscopic ancillary procedures (operative endoscopy) were attempted in eight patients, with a median of five procedures per patient (IQR 4–7). In one case, the endoscopy was only diagnostic. However, endoscopy was associated with dismal results (Table 4). Fistula closure with over-the-scope clipping (OTSC) was attempted in four patients; it failed in all cases. 

Along with the effluent control phase, the patient’s hypercatabolic state was counteracted by appropriately balanced nutritional support after nutritional therapy consultation. Parenteral nutrition (PN) was used to guarantee early nutritional support in all cases. The enteral route (per os (PO), enteral nutrition (EN), or fistuloclysis) was used in 93.75% cases. A successful transition to either EN or PO intake was possible in 62.55% cases, while fistuloclysis was performed in 31.25% cases.

Due to severe systemic compromise due to CEF itself and wound complications, 14 (87.5%) patients spent a relevant part of their hospital stay in the intensive care unit (ICU). The median number of ICU accesses per patient was 2 (IQR 1–3), with a median length of stay (LOS) per ICU access of 29 days (IQR 11.7–34.3 days) and an overall median ICU LOS of 71 days (IQR 28–101 days).

### 3.3. Surgical Rescue Phase 

#### Timeline to Surgery, Technical Planning, and FTR Rate 

Management of CEF is a time-consuming process. Patients who achieved spontaneous fistula closure were discharged home after a median LOS of 232 days (IQR 158–234 days) with home physiotherapy support.

Four of seven patients in whom the CEF was converted to a simple stoma did not heal with non-operative management. The conversion to a simple stoma allowed us to defer definitive surgery for several months. Patients were discharged from the hospital after a median LOS of 101 days (IQR 85.2–107.7 days) and readmitted after a median time of 140.5 days (IQR 73–208.5 days) for definitive surgery. Three patients required a period of rehabilitation before definitive take-down. Home nursing care was activated when a single patient was discharged home.

The overall median time from Niguarda admission to definitive surgery in 13 patients who required surgical removal of the fistula was 120 days (IQR 34–231 days).

For four patients who underwent ostomy, the median time from the first admission to definitive surgery was 236 days (IQR 148.5–328.5 days).

However, for nine patients in whom conversion of the CEF to a simple stoma not possible, definitive surgery was performed during the initial hospitalization after a median time from admission of 34 days (IQR 16–120 days).

Definitive surgery was performed only after a sound understanding of the fistula anatomy and planning of abdominal wall reconstruction and after the patients were free from sepsis (Figure 1) and had a good health status, as indicated by a median P-Possum Score of 20 (IQR 16–23). Before the skin incision, all fistulas were cannulated with a Foley catheter to prevent enteric spillage in the surgical field and identify the bowel loop during the surgical procedure. In the case of EAF, the skin incision was performed using a lateral surgical approach via the circumference of the granulation tissue containing the fistulas. Once the fistulised tract was resected, intestinal continuity was restored in 12 of 13 (92.3%) patients, with a median number of anastomoses per patient of 1 (IQR 1–2). All anastomoses were hand-sewn. Terminal sutures were preferred. One definitive stoma was performed. While attempting fascial closure, intra-abdominal pressure (IAP) was recorded to avoid the onset of abdominal compartment syndrome. 

In six (46.1%) patients, the anterior component separation technique was synchronously adopted to achieve fascia closure. In these patients and four patients for whom the risk of infection was considered high after primary fascial closure without component separation (76.9% cases), suprafascial NPWT was adopted.

Prostheses were necessary to achieve abdominal wall reconstruction in five (38% cases). In three patients, an inlay non-crosslinked biological porcine-derived mesh was used, while in one patient, a sandwich technique was applied, with a synthetic mesh overlapped with the biological mesh. A sub lay absorbable mesh was used only in one patient.

Fistula recurrence occurred in five patients. Complications requiring postoperative revision occurred in 69.2% cases. Seven and three surgical revisions were performed for bowel perforation and anastomotic failure, respectively.

The median number of postoperative revisions was two (IQR 1–5). Four patients in the recurrence group required surgical revision due to intraperitoneal and superficial uncontrolled fistula output, while one patient was treated with bedside procedures, and the fistula was converted to a simple stoma. 

Nine patients developed organ/space surgical site infection, and eleven patients developed superficial or deep surgical site infections. Fourteen patients developed sepsis due to catheter-related bloodstream infection (CRBSI). Among them, 13 patients experienced at least one episode of septic shock due to CRSBI. Moreover, four patients experienced at least one episode of hypovolemic shock. Three patients experienced acute hepatic failure (AHF). Fourteen patients required ICU support for at least one episode of organ dysfunction. The overall effective hospital LOS was 117 days (IQR 69.5–188.8 days).

The FTR rate was 25%. Three patients died of multiorgan failure (MOF). One patient, a 17-year-old man, died of cardiac arrest after discharge as a consequence of a suicide attempt with caustic ingestion. 

Table 5 summarises the burden of complications of patients who could not to be rescued.

## 4. Discussion

Our study aimed to explore the impact of a multidisciplinary rescue strategy on the FTR rate and postoperative burden of care in patients with CEF.

At the end of the first decade of 2000, the first patient affected by CEF was referred to our centre from a peripheral low-volume hospital. At that time, our General Surgery–Trauma Team Unit was rapidly growing along with our experience in managing patients undergoing emergency surgery and those with trauma.

We performed fistulae removal and bowel continuity restorage 66 days after admission. The patient required 35 preoperative procedures for fistula control and 53 procedures for complications after the first surgery. 

The patient died 305 days after the first surgical attempt, after 205 days spent in the ICU and an escalation of surgical and medical complications.

Since then, we have progressively refined our approach to patients with CEF, switching from basic surgical rescue to a more complex and multidisciplinary rescue strategy. 

During the last 10 years, we applied three essential phases for the management of such patients (Figure 1): Study phase: centralization of care, clinical history, and patients’ study;Sepsis control and strategy definition phase: bedside dressing, endoscopic approach, and metabolic support;Surgical rescue phase: timeline to surgery, technical planning, and FTR rate.

All patients in our case series were referred from low-volume hospitals distributed throughout Italy. We implemented a systematic and standardised three-step rescue strategy with high costs in terms of resources and time. We were able to rescue 11 patients, and among the five patients who experienced recurrence, four could not be rescued.

Management of CEF is particularly challenging and time consuming, and the burden of care can be overwhelming. Trauma and emergency surgeons should be aware that such patients’ clinical and surgical complexity can be confounding factors, limiting the potential opportunity for surgical rescue. We identified three essential phases of CEF management.

We have discussed the evidence analysing each phase of the surgical rescue strategy, exploring the impact of a multidisciplinary rescue strategy on the FTR rate and postoperative burden of care.

### 4.1. Study Phase 

#### Centralization of Care, Clinical History, and Patients’ Study 

All patients were centralised from peripheral hospitals for the treatment of CEF as a complex post-surgical complication. Patients were referred from all over the country, not only from hospitals within the Milan metropolitan area.

Our centre is the referral centre for patients with trauma and those undergoing acute care surgery in the Milan metropolitan area, with a volume of >700 trauma admissions per year, 30% patients with Injury Severity Score >16, and an average of 1000 emergency surgery procedures per year. Our centre is also a teaching hospital within the network of the University of Milan general surgery residency.

The importance of hospital volumes and surgeons’ experience in managing complications has been explored in recent studies. A review by Hatchimonji et al. focused on FTR in patients undergoing surgery, highlighting the essential aspects of ACS. Although the complication rate is not different between low- and high-volume hospitals, there is a dramatic difference in FTR rates in favour of high-volume hospitals. The same evidence was reported in a review by Zago et al. Acute care surgeons in referral centres have higher non-technical skills and the ability to coordinate with multidisciplinary teams [3,12].

Other studies have focused on the impact of surgeons’ experience, showing a much more significant effect on FTR rates compared to hospital volume. A persistent commitment to emergency surgery guarantees a higher performance and a more solid experience. Our hospital is a teaching centre for trauma and emergency surgery. A review by Hatchimonji et al. showed that teaching status could guarantee high levels of care. Some confounding factors, such as the availability of higher-quality resources, should be considered for teaching hospitals [12,15,16].

Once the patient is referred to our centre, we perform a systematic, thorough study of the fistula anatomy. The goal is two-fold—to evaluate the length of the bowel potentially available for an enteral route (PO, EN, or fistuloclysis) for nutrition and plan definitive intestinal reconstruction. The assessment should be performed with a combination of contrast radiological investigations (small bowel follow-up fluoroscopic examination, computed tomography [CT] with fistulogram, contrast injection into the fistula, and magnetic resonance imaging). The assessment should be performed using a combination of contrast radiological investigations or endoscopy [17]. In our series, all patients except two underwent both small bowel follow-up fluoroscopic examination and CT with fistulogram.

Along with the radiological study, we performed a deep study of clinical and surgical histories by retrieving patient records from the original hospital and speaking directly with previous surgeons.

The median number of fistulas in patients was two, making daily management more difficult. Di Saverio et al. reported that the number of fistulas and distance between openings are critical elements that can influence optimal wound care and definitive surgical management [18,19].

### 4.2. Sepsis Control and Strategy Definition Phase 

#### Bedside Dressing, Endoscopic Approach, and Metabolic Support 

The second phase of patient management defines the route toward spontaneous resolution or need for operative treatment.

Sound effluent control is paramount to minimise damage to surrounding tissues and convert a CEF to a simple enterocutaneous one manageable as a simple stoma. 

Dressing changes may be frequent for high-volume CEF, and bedside dressing changes may not always be feasible; hence, it may be performed in the OR. Moreover, the risk of iatrogenic damage to the bowel and surrounding tissues favouring the onset of new holes is relatively high; it is proportional to the number of dressing changes. In our series, the median number of procedures for CEF control performed in the OR per patient was 11 (IQR 2–33.5) [17,18].

Several techniques are available to optimise effluent control. We used a combination of different approaches, tailoring the choice of different devices to the characteristics of fistula (output volume and consistency), surrounding tissue, and surface of the granulation plate (flat, rough, friable, “beefy”). Among them, the nipple technique has been described and extensively applied in both trauma and non-trauma patients with CEF. Di Saverio et al. [19] described its application along with NPWT with the insertion of a Foley or Petzer catheter inside the nipple to collect fistula output. Attention should be paid to avoid damage to the soft granulation tissue or direct application to the exposed bowel [19,20,21]. In our series, the nipple technique was used in four patients with a proximal fistula and liquid effluent.

However, once the effluent was solid, the Fistula Adapter™ technique was applied to obtain effective spillage control. The Fistula Adapter™ technique is one of the most feasible solutions because of its easy application and limited damage to surrounding tissues. In addition, it allows the application of an ostomy bag over the device and can be used in conjunction with NPWT [22,23,24].

A recent case series of 13 patients treated at a level 1 U.S. trauma centre reported the successful application of the floating stoma technique to control the faecal output in seven of eight patients with no bowel damage [25]. In our series, the floating stoma technique was adopted in seven patients in an attempt to transform a deep fistula into a more superficial one. We did not use this technique for a prolonged time because of the risk of bowel damage by suturing the plastic drape to the fistula.

In our series, 13 patients were treated with NPWT in association with former dressing techniques to improve tissue granulation, which was successful in converting the CEF to a simple stoma in approximately 50% cases. Although converting the fistula with a well-nourished tissue using NPWT is the most effective approach to allow spillage control, it does not guarantee that the fistula will close. Some studies [26,27] have reported promising results with the use of NPWT, with a high rate of spontaneous closure providing a low fistula output and absence of protruding mucosa. In our series, this occurred in three patients in whom the fistula healed spontaneously. NPWT also reduced nurse and physician care, facilitating dressing changes since the system was replaced every 48–72 h. It also facilitated patient mobilization when feasible.

Eight patients needed an integrative endoscopic approach with long-term dismal results using either endoluminal stents, biliary stents, or over-the-scope clips.

Endoluminal stent placement was successful in only one case, characterised by a lateral jejunoileal fistula with no distal obstruction, in agreement with the findings of Rebibo et al. [28]. In other cases, endo-leaks between the stent and bowel wall and stent migration were observed. The failure of biliary stents in controlling fistula output may be an expression of complex high-output fistulas.

OTSC was associated with a 100% failure rate. Endoscopic intervention with OTSC represents an evolution in CEF treatment, which may broadly impact clinical practice. The technical profile of OTSC applications can be challenging in terms of localization of the fistula, navigation of the endoscope with the cap in place, and incorporation of chronically inflamed and scarring tissue. This last entity seems to play a pivotal role in the unfavourable results. Several studies [29,30,31] have demonstrated that the characteristic of chronic fistula (>30 days old) leads to reduced clinical success of OTSC in such a setting. Despite the technical success of 90% with OTSC applications, Roy et al. [30] obtained an overall clinical success rate of 70%, which decreased to 33% in chronic fistulas. In our series, all fistulas managed by OTSC were chronic and located in the jejunum or ileum. Despite the endoscopy skills available at our institution, these aspects may represent the reasons for OTSC failure in our series.

These endoscopic procedures should be used to gain control of CEF in highly selected cases and should be considered a bridge to surgery rather than a definitive solution [32]. 

Another distinctive characteristic of our series is the prolonged duration of hospital and ICU stay. The median LOS for spontaneous fistula healing and fistula conversion to a simple stoma was of 232 days (IQR 158–234 days) and 101 days (IQR 85.2–107.7 days), respectively. For patients who required at least one ICU access, the median ICU LOS per access was 29 days (IQR 11.7–34.3 days), while the overall ICU LOS was 71 days (IQR 28–101 days). These results are consistent with those reported by Teixeira et al. [33], indicating that the development of enterocutaneous fistula after trauma laparotomy was associated with a significant increase in ICU LOS and hospital LOS. High-quality ICU facilities and experienced critical care physicians are essential to support critically ill patients with CEF. Non-technical skills of acute care surgeons in coordinating patient management, regardless of the inpatient facility, are essential to reduce internal fragmentation of care. Healthcare workers involved in patient care must be aware of the importance of continuity of care and its positive impact on the outcome, especially of bedside procedures to control fistula output.

Determining the discharge destination is at the forefront of patient management. In our series, patients who achieved spontaneous healing were discharged home with physiotherapy support to improve motor function recovery after prolonged bed rest. Three of four patients in whom a simple stoma was achieved required inpatient rehabilitation prior to definitive surgery. The remaining patient was discharged home and supported by a home-nursing agency for stoma care. The most challenging aspect in patient discharge was aligning family and patient expectations with those of healthcare providers. This evidence supports the idea that discharge, when feasible, is achieved safely by addressing a patient’s clinical, psychological, social, and financial needs. 

### 4.3. Surgical Rescue Phase 

#### Timeline to Surgery, Technical Planning, and FTR Rate 

In patients not amenable to non-operative rescue, a surgical rescue attempt was planned. Definitive fistula removal should be performed when the patient’s condition is suitable and after sound understanding of the fistula anatomy. The abdominal wall reconstruction technique should be carefully planned.

Not all patients fulfil the metabolic and anatomical requirements to undergo surgery at the right time. Therefore, the duration of conservative treatment and timing of definitive surgery should be individualised according to patient characteristics. Fistula take-down may lead to disappointing results if performed too early due to dense adhesions, serosal lacerations, and mesenteric tears. Demetriades has reported that definitive surgery is ideally performed 4–6 months after fistula control in patients with CEF when peritoneal inflammation recovers [34]. Visschers et al. [35] suggested that patients should undergo surgery if spontaneous resolution is not achieved within 5–6 weeks. In their series of 79 patients, 49 required surgeries after a median time of 101 days (IQR 7–163 days). In our series, definitive surgery was performed after a median time of 120 days (IQR 34–231 days), similar to that suggested by Demetriades [34] and by Martinez et al. (median time 106 days; IQR 76–165 days) [36]. In patients undergoing ostomy, the median time to surgery was 236 days (IQR 148.5–328.5 days), longer than that suggested by Visschers et al. [35]. This might be due to the need for a period of rehabilitation prior to a definitive surgical attempt in three patients. 

In nine patients, definitive surgery was deemed necessary within a median time from admission of 34 days (IQR 16–120 days) to counteract the worsening metabolic derangement due to high-output fistula and ineffective wound care.

The adopted surgical technique has been standardised and previously described [17]. It is mandatory to avoid further damage to the exposed viscera and facilitate resection of the involved bowel loops. With the lateral approach suggested by Marinis et al. [37], the skin can be dissected very precisely, providing an easy exposure of the peritoneal content. Insertion of a Foley catheter into the fistulas is a ploy that allows easier fistula tract identification.

In our series, definitive surgery was successful in 12 (92.30%) patients. The intestinal continuity was restored, when feasible, with the lowest possible number of hand-sewn termino–terminal anastomoses. The median number of anastomoses per patient was one (IQR 1–2). We followed another study demonstrating that performing more than one anastomosis is a significant predictor of fistula recurrence [36]. Hand-sewn anastomoses were preferred over stapled anastomoses because evidence shows that anastomotic failures are more than twice as likely with stapled anastomoses than with hand-sewn anastomoses in case of friable or repeatedly manipulated bowel tissue, even when controlling for markers of preoperative nutrition [38]. Termino–terminal anastomoses were preferred for maximizing the absorptive surface. However, when a substantial difference in calibre between the loops was observed, side-to-side anastomosis was the primary choice [17].

Abdominal wall reconstruction can be challenging due to fascial retraction induced by the long time elapsed since the index procedure. There is no method of choice, but the ideal solution must fit the size of the abdominal wall defect, patient’s status, surgeon’s expertise, and available devices [17]. Regardless of the surgical technique, abdominal wall repair is crucial for monitoring IAP to prevent intra-abdominal hypertension. If IAP increases to >15 mmHg, the primary fascial closure attempt should be abandoned, and alternative strategies should be applied. In this setting, the techniques for component separation may generate >10 cm of the fascial plane at the midline of each side [17,39,40]. NPWT should be left in place for 48–72 h in cases of large dissections or those with a high risk of infections. In our series, the anterior component separation technique was successfully adopted in six patients in conjunction with NPWT to prevent seroma. NPWT was also used in four patients in whom primary fascial closure was achieved without component separation but at a high risk of infection. If component separation is insufficient to reapproximate the fascial edges, biological or biosynthetic meshes can be used as a bridge (inlay position). The use of non-cross-linked biological or biosynthetic prostheses should be advocated in all patients treated for CEF due to contamination of the surgical field [17,36,40]. The disadvantages of non-cross-linked meshes positioned inlay include a 40–50% rate of recurrent hernia and abdominal wall laxity [17]. Biosynthetic meshes seem promising, mostly because they are cost-effective, but there are limited data pertaining to their use in this setting [41].

If abdominal wall defects are >200 cm^2^, rotational or free flaps should be considered [42]. In our series, we used non-cross-linked biological meshes in the inlay position in three patients. In another case, the biological mesh was reinforced with an overlapped synthetic prosthesis (sandwich technique). In one patient in whom fascial closure was achieved, a retromuscolar (sub lay) absorbable mesh was used. Flaps were not necessary in any case.

Four patients could not be rescued. Two studies on the management of CEF described lower FTR rates and their relationship with clinical factors [23,33]. Wainstein et al. [23] reported a 6% mortality rate in a monocentric cohort of 62 patients. In their study, the factors related to death were preoperative hypoalbuminemia and more than two anastomoses. Martinez et al. [33] reported a mortality rate of 13%. In their study, fistula recurrence was the only factor related to death. 

Three patients in our series died of MOF after septic shock. In two patients, acute hepatic failure was the leading cause. All patients in our series were supported with long-standing parenteral nutrition. The occurrence of parenteral nutrition-associated liver disease [28] as an underlying factor in patients who develop hepatic failure cannot be excluded. Sepsis is a synergic factor in the pathogenesis of AHF.

The fourth patient who could not be rescued developed a severe depressive disorder secondary to a long-standing period of bedside procedures and failure of surgical rescue due to fistula recurrence. During hospitalization, the patient was supported by a psychologist and psychiatrist, similar to all other patients in the cohort. Despite the support, the patient refused further therapy or consultation after discharge and experienced a severe rebound. One month after discharge, he committed suicide via caustic ingestion and died from subsequent cardiac arrest. This event made us realise that the mental health of patients with CEF should not be underestimated.

The inability to control ongoing psychological disorders and the related consequences of daily care should be considered in the burden of complications contributing to the FTR rate. Compliance with bedside dressing is an essential factor that contributes to fistula control. Patients should undergo a structured empowerment process to understand the importance of daily cooperation and visualise a common care target.

Almost 70% patients required a median of two revisions for postoperative complications. All patients who could not be rescued experienced an escalation of surgical and medical complications as shown in Table 5.

Hatchimonji et al. discussed the impact of multiple complications on the FTR rate, leading to a paradigm shift in the definition of FTR in general emergency surgery. The traditional definition of FTR accounts for only one complication. Wakeham et al. hypothesised that in patients undergoing surgery, death occurs after a cascade of complications. The authors demonstrated that the risk of developing a complication was higher in patients with a history of a previous event. Both events were connected, indicating that a second complication was the consequence and evolution of the index event [43]. 

Hatchimonji et al. identified the number of complications as an independent risk factor for FTR in general emergency surgery. From one to six complications, the risk of FTR increases two- to eight-fold [12]. 

The definition of the FTR rate is not universal and relies on the nature of the population considered. Different definitions have been proposed [3,12]. 

Patients with CEF represent a definite group in which a wide pattern of postoperative events can influence the FTR rate. The clinical management of CEF follows the natural history of the disease and attempts to control the unpredictable evolution of a surgical nightmare.

Patients with CEF may require a tailored definition of FTR. The factors influencing clinical evolution are both surgical and medical. The burden of complications, regarded as the effective number of complications and the procedures to deal with them, is also crucial. Moreover, the psychological impact of long-standing disease cannot be ignored. Patients with CEF require extremely long hospitalizations, multiple ICU accesses, prolonged artificial nutrition support, and bed rest and experience frequent failures of bedside procedures for fistula control and daily dressing change, contributing to the worsening of their underlying depressive disorders.

Considering previously published literature and our experience, the following complications may be considered to better define and understand the route to FTR in patients with CEF: fistula recurrence, shock, sepsis, colonization from multidrug-resistant microorganisms, organ/space surgical site infections, unplanned postoperative revisions, hepatic failure, pneumonia, deep venous thrombosis/pulmonary embolism, uncontrolled depressive disorders, and low patient empowerment [2,3,12]. 

This study has several limitations, including its retrospective design, the limited number of patients, and data loss due to the complex clinical and surgical histories of patients and their late centralization. 

However, our monocentric case series is not irrelevant given the rarity and complexity of the disease and its related burden of care. Our long-standing experience allowed us to refine and apply a standardised rescue strategy to understand the natural history and limit the unpredictability of patients with CEF.

## 5. Conclusions

Our experience, along with published data, should open and stimulate a debate on the need for tailored definitions of FTR in patients undergoing emergency general surgery considering specific diseases, especially in those affected by CEF.

Such patients represent a unique cluster and a continuous challenge in which medical, surgical, and psychological factors can be real game changers. 

Implementing a multicentre registry or trial to better understand the natural history of CEF and define factors influencing rescue or FTR is desirable.

## Figures and Tables

**Figure 1 jpm-12-00292-f001:**
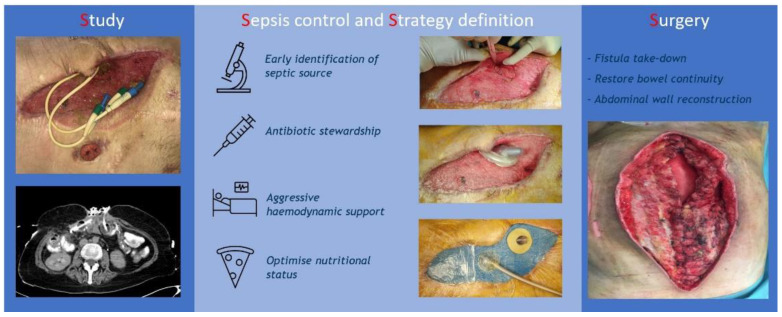
Three-step standardised approach to CEF.

**Table 1 jpm-12-00292-t001:** Clinical and surgical history until enteric fistula onset.

Patient	Sex	Age (Years)	ASA	BMI	Past Medical History	Past Surgical History	Procedures	Open Abdomen	Time to EF (Days)	Dead	/Recurred
1	Male	59	2	31	Hepatic cirrhosis	Bowel ischemia due to portomesenteric thrombosis	5	No	25	No	No
2	Male	17	3	22	MCTD-Megaoesophagus	Gastric perforation	6	Yes	19	Yes	No
3	Male	45	2	27	IM	Bowel perforation following bowel obstruction	6	Yes	47	No	No
4	Male	32	3	29	BWS syndrome	Bowel perforation following bowel obstruction	2	No	17	No	No
5	Female	52	2	26	None	Traumatic bowel perforation	3	Yes	28	No	No
6	Female	77	3	25	Rectal and endometrial cancer	Bowel iatrogenic perforation after hernia repair	3	Yes	24	No	No
7	Male	61	2	27	None	Bowel iatrogenic perforation after lap cholecystectomy	5	Yes	30	No	No
8	Female	36	2	32	Obesity, RYGB	Bowel ischemia due to internal hernia	6	Yes	17	No	Yes
9	Female	73	2	26	None	Bowel iatrogenic perforation after hernia repair	2	No	40	No	No
10	Female	61	3	24	Gastric cancer	Esophago-jejunostomy leakage	7	No	33	No	No
11	Male	47	2	33	Obesity, heavy smoker	Bowel perforation following acute mesenteric ischemia	5	Yes	70	No	No
12	Male	46	2	26	TTP, PNH	Bowel perforation following acute sesenteric ischemia	8	Yes	53	No	No
13	Male	36	3	25	CDH, IM	Bowel perforation following bowel obstruction	1	No	15	No	No
14	Female	51	2	30	Obesity, LSG	Bowel iatrogenic perforation after lap sleeve gastrectomy	6	Yes	18	Yes	Yes
15	Female	41	3	27	IM, Duodenal stenosis	Bowel perforation following bowel obstruction	22	No	180	Yes	Yes
16	Male	20	2	23	None	Bowel perforation following appendectomy	13	No	145	Yes	Yes

Abbreviations: ASA American Society of Anaesthesiology Score; BMI Body Mass Index, MCTD, mixed connective tissue disease; IM, intestinal malrotation; BWS, Beckwith–Wiedemann syndrome; RYGB, Roux-en-Y gastric bypass; TTP, thrombotic thrombocytopenic purpura; PNH, paroxysmal nocturnal haemoglobinuria; CDH, congenital diaphragm herniation; LSG, laparoscopic sleeve gastrectomy.

**Table 2 jpm-12-00292-t002:** Investigation phase. Fistula location for each patient and type of radiological study are shown.

Patient	Fistula Location				Imaging	
	Duodenal	Jejunal	Ileal	Colic	CT Scan	GI X-ray
1		●	●		●	●
2	●	●	●	●	●	●
3		●			●	●
4		●			●	●
5	·	●	●	·	●	●
6		●		●	●	●
7			●		●	●
8			●		●	
9	·		●		●	●
10			●			●
11			●		●	●
12		●	●		●	●
13			●		●	●
14		●	●		●	●
15		●			●	●
16		●			●	●

**Table 3 jpm-12-00292-t003:** Fistula features and techniques for output control.

Patient	Fistula Features					Effluent Control				
	Deep	Superficial	Low-Output	Medium-Output	High-Output	NPWT	Nipple	Fistula Adapter	Floating Stoma	Skin Graft
1		•	•							
2	•	•			•	•	•			•
3		•			•	•		•		
4		•		•		•	•	•	•	•
5	•	•			•	•	•		•	•
6		•			•	•			•	
7		•	•			•			•	•
8	•		•							
9	•				•					
10		•	•							
11	•			•		•	•		•	
13	•				•				•	
14		•	•						•	
15		•			•	•		•		
16		•			•	•		•		•

NPTW, negative pressure wound therapy.

**Table 4 jpm-12-00292-t004:** Summary of endoscopic management of CEF patients. OTSC, over-the-scope clip.

Patient	Operative Endoscopy	Procedures	Biliary Stent	Endoluminal Prosthesis	OTSC	Endoscopic Healing
1	•	1	•		•	
2	•	7	•	Duodenal		
3	•	7	•	Ileo-colic anastomosis	•	
4		0				
5	•	1		Duodenal-jejunal anastomosis		
6		0				
7		0				
8		0				
9	•	5	•	Duodenal		
10		0				
11		0				
12		0				
13	•	1			•	
14	•	1		Jejunum-ileal anastomosis		•
15	•	1			•	
16		0				

**Table 5 jpm-12-00292-t005:** Complications in patients who failed-to-be-rescued.

Pts	Sex	Age (Years)	Postoperative Surgical Revisions	Recurrence	Bowel Perforation	Anastomotic Failure	Organ Space SSI	Shock	AHF	MDR	Depressive Disorder	ICU	FTR
2	M	17	18	•			•	•			•	1	•
14	F	51	7	•		•	•	•	•	•	•	4	•
15	F	41	1	•	•		•	•	•	•	•	3	•
16	M	20	53	•	•	•	•	•		•		4	•

SSI, surgical site infection; AHF, acute hepatic failure; MDR, multidrug-resistant microorganisms; ICU, intensive care unit access; FTR, failure to rescue.

## Data Availability

The data presented in this study are available on request from the corresponding author. The data are not publicly available due to privacy matters.

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
