# Peer review of "Failure to Rescue (FTR) and Pitfalls in the Management of Complex Enteric Fistulas (CEF): From Rescue Surgery to Rescue Strategy"

_jpm, 2022, doi:10.3390/jpm12020292_

Round 1

Reviewer 1 Report

Thanks to the authors for submitting their article for evaluation in our journal. The article is interesting because the treatment of complex fistulas is a challenge. The work is well structured, the design is correct and the results are of interest. We suggest some changes to improve the quality of it:

  1. 1.- The introduction is very well written, its reading is easily understandable. Please include the meaning of FTR the first time it appears in the text at line 67 on page 2.
  2. Although the definition of complex fistula is described in the introduction to the article, please include material and methods as inclusion and exclusion criteria
  3. Please, define OTSC in the Table 4 heading.
  4. The conclusion is too long, it should be shortened
  5. Rewrite the references according to Author´s Guidelines.
  6. Rewrite Ethical Statment according to Author´s Guidelines.

Author Response

  1. - The introduction is very well written, its reading is easily understandable. Please include the meaning of FTR the first time it appears in the text at line 67 on page 2.

Thanks for the suggestion. I included the meaning.

  1. Although the definition of complex fistula is described in the introduction to the article, please include material and methods as inclusion and exclusion criteria

We integrated the definition and better specified inclusion and exclusion criteria in the methods section.

  1. Please, define OTSC in the Table 4 heading.

The acronym was specified in the Table 4 heading.

  1. The conclusion is too long, it should be shortened

Thanks for the advice, we followed your suggestion shortening the conclusion.

  1. Rewrite the references according to Author´s Guidelines.

References were corrected and reported following journal guidelines.

  1. Rewrite Ethical Statment according to Author´s Guidelines.

The sentence was modified as requested.

Reviewer 2 Report

The research design should be recognised in the abstract. There is ambiguity around the research methodology. Is this a cross-sectional study? Cohort or experimental research?

Line 67, FTR has not been defined in the full name.

Please add a beckground on previous research on this topic before the methods section.

The research methods should be started with the design and related methodological aspect.

Many details in the research methods are missing. Elaboration for all aspects are needed to make it possible to judge its quality: Sample and settings, recruitment process, data collection tools, data analysis process, ethical considerations etc.

I wonder about the type of the research method to evaluate the article based on reporting standards. You are advised to report your article based on the claim research method. The related methodologcal checklist can be found here:

Reporting guidelines | The EQUATOR Network (equator-network.org)

Author Response

Poin to point letter to reviewer 2

  1. The research design should be recognised in the abstract. There is ambiguity around the research methodology. Is this a cross-sectional study? Cohort or experimental research?

The study design was specified in the abstract and also in the methods section. Thanks for the advice.

  1. Line 67, FTR has not been defined in the full name.

The acronym was specified. Thanks.

  1. Please add a background on previous research on this topic before the methods section.

The background section was implemented as suggested. Thanks for your observation.

  1. The research methods should be started with the design and related methodological aspect.

The study design was defined in the methods section.

  1. Many details in the research methods are missing. Elaboration for all aspects are needed to make it possible to judge its quality: Sample and settings, recruitment process, data collection tools, data analysis process, ethical considerations etc.

Thanks for your comment. Methodological details were better specified in the methods section as suggested. In regard of ethical considerations, a statement can be found in the specific “backmatter” section.

  1. I wonder about the type of the research method to evaluate the article based on reporting standards. You are advised to report your article based on the claim research method. The related methodologcal checklist can be found here: Reporting guidelines | The EQUATOR Network (equator-network.org)

Thanks for your precious advice. As reported in the methods section, reference 14, we followed the STROBE statement checklist for reporting of observational studies, since this study is a retrospective observational one.

Round 2

Reviewer 2 Report

nothing more

Author Response

Best wishes 
